# Radiation Protection in Interventional Radiology/Cardiology—Is State-of-the-Art Equipment Used?

**DOI:** 10.3390/ijerph182413131

**Published:** 2021-12-13

**Authors:** Christiane Behr-Meenen, Heiner von Boetticher, Jan Felix Kersten, Albert Nienhaus

**Affiliations:** 1Competence Centre for Epidemiology and Health Services Research for Healthcare Professionals (CVcare), Institute for Health Services Research in Dermatology and Nursing (IVDP), University Medical Centre Hamburg-Eppendorf (UKE), 20246 Hamburg, Germany; j.kersten@uke.de (J.F.K.); albert.nienhaus@bgw-online.de (A.N.); 2Division for Medical Radiation Physics, Faculty VI: Medicine and Health Sciences, Carl von Ossietzky University Oldenburg, 26121 Oldenburg, Germany; heiner.vonboetticher@web.de; 3Department of Occupational Medicine, Toxic Substances, Health Service Research, German Statuary Institution for Accident Insurance and Prevention for Health and Welfare Services (BGW), 22089 Hamburg, Germany

**Keywords:** occupational radiation exposure, interventional medicine, occupational medicine, interventional installations, ceiling-suspended lead acrylic shield, personal protective equipment, dosimetry

## Abstract

Interventional radiology/cardiology is one of the fields with the highest radiation doses for workers. For this reason, the International Commission on Radiological Protection (ICRP) published new recommendations in 2018 to shield staff from radiation. This study sets out to establish the extent to which these recommendations are observed in Germany. For the study, areas were selected which are known to have relatively high radiation exposure along with good conditions for radiological protection—interventional cardiology, radiology and vascular surgery. The study was advertised with the aid of an information flyer which was distributed via organisations including the German Cardiac Society (Deutsche Gesellschaft für Kardiologie- Herz- und Kreislaufforschung e. V.). Everyone who participated in our study received a questionnaire to record their occupational medical history, dosimetry, working practices, existing interventional installations and personal protective equipment. The results were compared with international recommendations, especially those of the ICRP, based on state-of-the-art equipment. A total of 104 respondents from eight German clinics took part in the survey. Four participants had been medically diagnosed with cataracts. None of the participants had previously worn an additional dosimeter over their apron to determine partial-body doses. The interventional installations recommended by the ICRP have not been fitted in all examination rooms and, where they have been put in place, they are not always used consistently. Just 31 participants (36.6%) stated that they “always” wore protective lead glasses or a visor. This study revealed considerable deficits in radiological protection—especially in connection with shielding measures and dosimetric practices pertaining to the head and neck—during a range of interventions. Examination rooms without the recommended interventional installations should be upgraded in the future. According to the principle of dose minimization, there is considerable potential for improving radiation protection. Temporary measurements should be taken over the apron to determine the organ-specific equivalent dose to the lens of the eye and the head.

## 1. Introduction

In interventional radiology/cardiology, workers may be exposed to radiation. Along with an increase in interventional procedures, the literature cites lax usage of interventional installations and unconscientious use of personal protective equipment as possible reasons for radiation exposure [1,2,3,4,5,6,7]. The International Commission on Radiological Protection (ICRP) published practical advice in 2018 to protect all workers involved in interventions. This advice includes exposure monitoring strategies and training along with recommendations on protective garments and interventional installations [8].

The ICRP recommends wearing one dosimeter underneath the lead apron and another one over it. It emphasises that the best way to assess the organ-specific equivalent dose to the lens of the eye and the head in real operating conditions is by wearing a dosimeter over the lead apron [8,9].

All staff who are present in the X-ray room during the procedure should wear personal protective equipment (PPE) consisting of an apron and thyroid protection. Lead equivalences of 0.35 or 0.5 mm are customary for aprons, while 0.5 mm is standard for thyroid protection. As PPE does not shield the whole body from scatter radiation, the head and eyes in particular are unprotected [8,9,10,11].

According to the ICRP, the consistent and correct use of a ceiling-suspended lead acrylic shield is the best way to protect the head and eyes [3,11,12]. Consistently wearing lead glasses is another important means of protecting the lens of the eye from scatter radiation [8]. Several authors also recommend wearing radiation-absorbing surgical caps or headbands to protect the brain [13,14].

The ICRP points out that, if the X-ray tube is positioned underneath the examination table, it is relatively easy to shield the primary beam by hanging a drape with a lead equivalence of 0.5 mm under the table on both sides [15,16,17,18]. For this reason, it recommends positioning the X-ray tube underneath the table. The ICRP states that the hands of interventional medical practitioners can be protected by a folding, table-mounted shield. According to the ICRP, the use of a patient apron with a lead equivalence of 0.5 to 1.0 mm reduces scatter radiation and therefore protects the physicians’ hands as well. The use of a patient apron has to be placed outside the field of the primary beam [19].

To reduce both patient and staff radiation exposure, the ICRP recommends minimising the fluoroscopy time, image frequency, and number of images per examination [8].

The objective of this study is to investigate the extent to which international recommendations, particularly those contained in the 2018 ICRP Publication 139—such as those to protect the head and neck—are observed in interventional medicine.

## 2. Materials and Methods

For the study, we selected areas which are known to have relatively high radiation exposure along with good conditions for radiological protection—interventional cardiology, radiology and vascular surgery. The study was advertised with the aid of an informational flyer which was distributed via organisations including the German Cardiac Society (Deutsche Gesellschaft für Kardiologie-Herz- und Kreislaufforschung e. V.). After they were each informed about the content of the study, all participants provided written consent. They received a personal ID number and a questionnaire to record their occupational medical history and any parameters which could affect the dosage figures, such as surgical techniques, the type and frequency of radiological/interventional procedures, the interventional installations used, the technical features of the radiation source, and PPE.

We also asked specific questions about dosimetry, such as the method of measurement, the use of partial-body dosimeters, the position in which the dosimeter is worn and annual doses. For the occupational group of cardiologists, details of the fluoroscopy time, image frequency, and the number of images per examination were also assessed. The results of the survey were compared with the ICRP’s recommendations on protection from occupational radiation exposure during interventional procedures. The statistical data analysis was performed using the software tool IBM SPSS 26 and version 4.0.3 of the statistical evaluation program R.

Absolute figures and ratios for categorical data are provided along with e.g., averages and medians for numeric variables. We used the Kruskal-Wallis test to analyse whether there is a significant difference between the results for the various clinics. The data was collected in accordance with ethical, data protection, and professional standards and requirements. An occupational ethical and legal consultation was conducted with the Ethics Commission of the Hamburg Medical Association and was assigned case number PV7216.

## 3. Results

A total of 104 respondents from eight German clinics took part in the survey. Of the workers contacted personally, 90% decided to participate. The study participants were people who experienced occupational exposure to radiation and who were in the immediate vicinity of a radiation source—usually to the side of the X-ray table—during medical interventions involving fluoroscopy. Of the participating physicians, eleven worked in interventional radiology, 57 in interventional cardiology and four in vascular surgery. 32 assistants working in interventional cardiology and vascular surgery also completed the survey. The majority of the study participants were male (67.3%), cardiologists (54.8%) and had been working in interventional radiology/cardiology for less than ten years (61.4%; Table 1).

Of 101 valid responses, four respondents answered “yes” to the question of whether they had been medically diagnosed with a cataract. This corresponds to 4% of all participants and 10.3% of those who had been exposed to ionising radiation for at least ten years. All of the participants stated that they wore the official dosimeter underneath their protective garments and did not normally wear an additional dosimeter over their lead apron. Not a single clinic had acted on the recommendation to wear an additional dosimeter over the lead apron, as that is the best way to assess the organ-specific equivalent dose to the lens of the eye and the head in real operating conditions. At the time of the survey, just two doctors had ever used an eye dosimeter during their work. As recommended by the ICRP, all of the participants stated that they wore “radiation protection garments” in the examination room. Table 2 summarises the information provided about the different lead equivalences of the radiation protection garments used. In our study, 99 respondents stated that they “always” wore thyroid protection. Just two participants reported that they only wore thyroid protection “frequently” (Table 2). Responses to the questions about use of lead glasses or visors revealed a mixed picture. Glasses with a lead equivalence of 0.5 mm were used by 45 participants.

Meanwhile, 14 respondents wore glasses with a lead equivalence of 0.75 mm. Of the study participants, 31 stated that they “always” wore their lead glasses or visor and 16 wore them “frequently” (Table 2).

With regard to how often lead glasses were worn, there was no statistically significant difference between the clinics when all four categories (“always”; “frequently”; “rarely”; “never”) were taken into account (Figure 1).

However, conducting the test with two combined categories (“always/frequently” and “rarely/never”) results in a *p*-value of *p* = 0.035 after dichotomisation. This reveals a statistically significant difference in the use of lead glasses at the individual clinics. When asked “How often do you wear a cap?” or “How often do you wear a headband?” 14 replied “always”, five stated “frequently”, and ten answered “rarely”. A total of 70 participants reported that they “never” wore a cap or a headband with a protective lead equivalence (Table 2). Of the respondents, 95 stated that the examination rooms were fitted with a ceiling-suspended lead acrylic shield. Just four participants replied that there was no ceiling-suspended lead acrylic shield in the operating room. When asked “How often do you use a ceiling-suspended lead acrylic shield?” 78 participants replied “always” and 18 responded with “frequently” (Table 3).

In the experience of 90 respondents, the X-ray tube is always underneath the table, as shown in Figure 2. Just nine respondents stated that it was sometimes (for 2% to 25% of interventions) or predominantly (for 70% to 95% of interventions) located above the table. Two of the physicians surveyed replied that the X-ray tube had been above the table in 100% of the interventions conducted by them (Figure 2).

When asked “Is there a drape under the examination table?” 31 of the study respondents replied that there was a drape on both sides. However, 67 participants stated that there was only a drape on one side and 26 reported that there was no drape under the table at all (Table 3). A folding, table-mounted shield was fitted to the examination table according to 84 of the respondents. A total of 61 study participants reported that they “always” used the table-mounted shield, while a further 20 confirmed that it was “frequently” utilised (Table 3).

When asked “How often is a patient apron used?” 30 participants replied “always” and 33 responded with “frequently”. Meanwhile, 25 people stated that they “rarely” or “never” used a patient apron (Table 4). 

In our survey, 49 participants responded that they always used all the technical means of reducing radiation. A further 45 respondents answered this question with “frequently”. The question “Are there particular working practices that you believe reduce radiation?” was answered by 13 participants. Respondents mentioned the so-called tiger catheter technique and using a long wire to change catheters. As additional technical means, the participants cited regular image fades, using a 3-D mapping technique for navigation, and FORS technology (Table 4). To reduce both patient and staff radiation exposure, the ICRP recommends minimizing the fluoroscopy time, image frequency, and number of images per examination. Table 5 shows the information provided by surveyed cardiologists on the parameters affecting exposure. The fluoroscopy time varies between one and 30 min. Meanwhile, the image frequency ranges from three to 15 images per second and the number of images falls somewhere between one and 20 per minute.

## 4. Discussion

Of all the participants in our study who had been exposed to ionising radiation for over ten years, four (10.3%) stated that they had been diagnosed with a cataract. By way of comparison, a prospective study from 2004 which surveyed a total of 35,705 medical-technical radiology assistants (MTRAs) found that 6.7% of them had developed a cataract after twenty years in the occupation [20].

Without suitable eye protection, employees with an average or high workload may undoubtedly exceed the new annual equivalent maximum dose for the lens of the eye of 20 mSv p.a. [8].

In a 2013 study of 295 people who had worn a second dosimeter over their apron, 53 were exposed to a personal surface dose H_p_(0.07) of over 20 mSv p.a. and a further 69 were exposed to over 10 mSv p.a. [21].

The results of this study showed that none of the participants wore an additional dosimeter over their lead apron, as recommended in other publications. To make it possible to determine the organ-specific equivalent dose to the lens of the eye and the head, measurements should be taken at least temporarily using a standard whole-body dosimeter worn over the apron [8,9].

Correct use of a ceiling-suspended lead acrylic shield can reduce scatter radiation by a factor of 2 to 10 [22,23]. A ceiling-suspended lead acrylic shield protects the head and neck as well as the eyes [24]. In our study, 95 participants stated that the examination rooms they used were fitted with ceiling-suspended lead acrylic shields. However, four respondents reported that no ceiling-suspended lead acrylic shield had been installed. Of the 104 study participants, 78 used the available ceiling-suspended lead acrylic shields “always”, 18 used them “frequently”, and one stated that the ceiling-suspended lead acrylic shield was rarely used. Especially when they are fitting pacemakers or defibrillators, doctors usually work very close to the radiation source and it can often be very difficult to use a ceiling-suspended lead acrylic shield. To optimise radiation protection, the corresponding examination rooms should be upgraded and staff should be trained and briefed so as to support the constant use of ceiling-suspended lead acrylic shields. Assistants can also be protected by means of mobile radiation shielding [8].

All measures which protect the patient from unnecessary ionising radiation simultaneously reduce the scatter radiation to staff, and therefore their occupational exposure. These include, for example, the fluoroscopy time, the number of images and the image frequency [8]. The cardiologists who took part in our study stated that the median of their typical fluoroscopy time was 5 min (1–30). Our results show that fluoroscopy times vary greatly, with a range of 29 min. This is doubtless due in part to the type of procedure, but also how experienced the individual physicians are. With regard to the typical image frequency, the median stood at 7.5 images per second in our study (3–15). Twenty years ago, the preferred standard image frequency was between 25 and 50 images per second [10]. This shows that new X-ray equipment makes it possible to decrease the image frequency used and to reduce the associated radiation exposure for staff and patients alike. Nevertheless, our results reveal that the image frequency used ranges from 3 to 15 images per second among the respondent cardiologists. This variation is almost certainly attributable to individuals’ customary working practices, but also to image quality requirements. Operators must be advised that they should use the lowest acceptable image quality wherever possible to enable them to reduce image frequency in the interests of radiation protection. The same applies to the number of images; the median here is six per examination. Here too, unnecessary images should be avoided with a view to radiation exposure.

According to the ICRP, scatter radiation can be reduced further by using patient aprons with a lead equivalence of 0.5 to 1.0 mm. However, it must be ensured that this apron is not positioned between the patient and the useful beam because this can automatically increase the radiation exposure for both the patient and the staff [8]. In our study, more than 70% of participants stated that a patient apron was used “always” or “frequently”. Meanwhile, 28.4% “rarely” or “never” used one. Given the results of the study, we recommend training and briefings to encourage greater use of patient aprons so as to reduce scatter radiation and the associated exposure to the upper body—especially the eyes, thyroid and head.

In our study, 70 respondents stated that they “never” wore a cap or a headband with a protective lead equivalence.

However, in the literature, a number of authors recommend wearing radiation protection caps or headbands [13,14]. Meanwhile, others claim that caps do not offer any protection due to the geometry of the ray path [25].

In certain cases where the radiation originates from below, this is no doubt correct. However, there are plenty of situations in which the radiation does not just come from below. Given this, wearing a protective cap or headband is undoubtedly a further, very important component of radiation protection in line with the principle of dose limitation. Especially during examinations and procedures where an above-table position is used and either fixed, leaded acrylic screens or mobile radiation shielding can only be utilised to a limited degree, operators’ PPE should be expanded with protective caps or headbands.

According to the ICRP, a folding, table-mounted shield can provide additional protection for the hands of interventional medical practitioners. Just 61 of the respondents stated that they “always” used these.

As recommended by the ICRP, the consistent use of lead glasses is another important means of protecting the lens of the eye from ionising radiation [8]. However, this cannot replace the use of a ceiling-suspended lead acrylic shield and should only be seen as a supplementary preventive measure. When the physician looks at the monitor, scatter radiation can reach the eye from below and from the side through the unprotected parts of the lead glasses. The organ-specific equivalent dose to the lens of the eye is therefore determined to a large degree by the operator’s position and the beam angle [8].

It is more important for lead glasses to fit well than to have a high lead equivalence so that they also shield radiation from below and the side [24]. In other studies, glasses with a lead equivalent of 0.5 mm combined with a large lens are recommended by interventional radiologists as a more adequate and effective protection of the eye lens [26].

However, theoretically, a lead equivalence of 0.75 mm can reduce scatter radiation by more than 85% [22,23,24].

Our findings show that just 30 participants always wear lead glasses (34.5%) or a visor (2.1%). Especially in situations where surgical techniques make it difficult to use a ceiling-suspended lead acrylic shield at all times, workers should ensure that they also wear lead glasses. This could substantially improve radiological protection in line with the principle of dose limitation. The results of the survey show that 90 (86.5%) of the 104 participants always work with the X-ray tube underneath the table. This position makes it relatively easy to shield the primary beam by hanging drapes under the table on both sides [15,16,17,18].

Using these curtains can reduce the radiation exposure to the legs by a factor of 10 to 20. However, less than a third of participants stated that drapes had been hung under the table on both sides. It is much more difficult to shield the primary beam in an above-table position, which is sometimes used to fit pacemakers or for electrophysiological examinations, for example. In this study, eleven respondents stated that the X-ray tube was sometimes, mainly or always positioned above the table. In these cases, it is not always possible to use ceiling-suspended lead acrylic shields consistently for all procedures, as mentioned above. Wearing lead glasses or visors is absolutely essential in these situations.

Generalizing the results could be problematic as the participating clinics were not selected at random. It is possible that only clinics that already had very good radiation protection took part in the study.

## 5. Summary

To optimise radiation protection, examination rooms which do not have fixed, ceiling-suspended lead acrylic shields should be upgraded. Consistent use of the screens should be supported by means of training and briefings. The same applies to usage of lead glasses or a visor.

Especially in situations where surgical techniques make it difficult to use a ceiling-suspended lead acrylic shield or mobile radiation shielding at all times, workers should ensure that they also wear lead glasses. This could substantially improve radiological protection in line with the principle of dose limitation. To make it possible to determine the organ-specific equivalent dose to the lens of the eye and the head, measurements should be taken at least temporarily over the apron.

## Figures and Tables

**Figure 1 ijerph-18-13131-f001:**
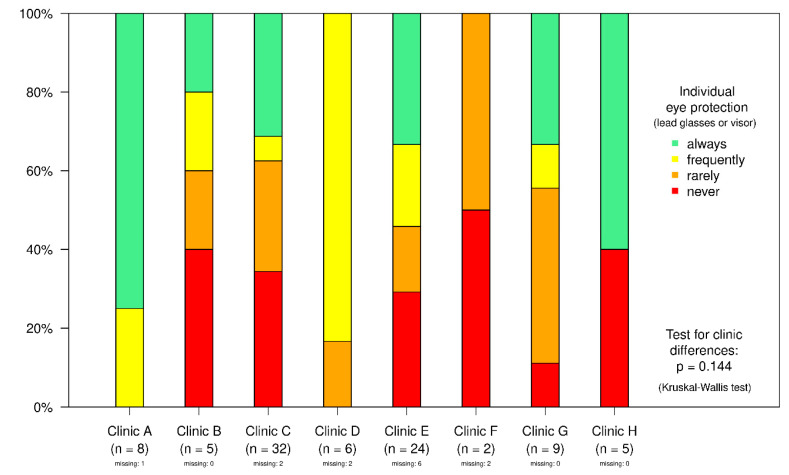
Use of lead glasses and visors.

**Figure 2 ijerph-18-13131-f002:**
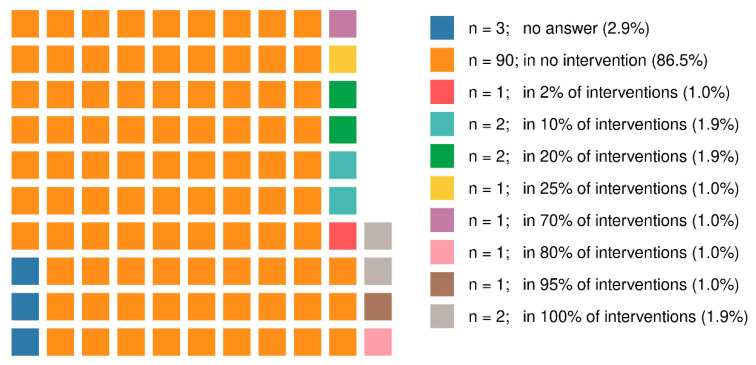
Location of the X-ray tube above the table; percentage of procedures (*n* = 104).

**Table 1 ijerph-18-13131-t001:** Description of study participants (N = 104).

Variable	N (%)
Gender *	
Women	33 (32.7%)
Men	68 (67.3%)
Age in years	
Mean ± SD (min.–max.)	43.1 ± 10.3 (23–65)
Specialism	
Interventional cardiology	57 (54.8%)
Interventional radiology	11 (10.6%)
Vascular surgery	4 (3.8%)
Assistants (cardiology and vascular medicine)	32 (30.8%)
Number of years working in interventional medicine *	
1 to 9	62 (61.4%)
10 to 19	20 (19.8%)
Over 20	19 (18.8%)

* Figures quoted as numbers and percentages are based on the valid data.

**Table 2 ijerph-18-13131-t002:** Information on personal protective equipment (N = 104).

Question *	Always	Frequently	Rarely	Never	N (%)
Which radiation protection garments do you wear?					
Lead equivalence 0.35 mm					54 (55.7)
Lead equivalence 0.5 mm					43 (44.3)
How often do you wear thyroid protection?	99 (98.0)	2 (2.0)			
Do you wear lead glasses?					
Yes, lead equivalence 0.5 mm					45 (76.3)
Yes, lead equivalence 0.75 mm					14 (23.7)
How often do you wear lead glasses?	30 (34.5)	15 (17.2)	19 (21.8)	23 (26.4)	
How often do you wear a visor?	1 (2.1)	1 (2.1)	10 (20.8)	36 (75.0)	
Do you wear a cap?					
Yes, lead equivalence 0.25 mm					10 (43.5)
Yes, lead equivalence 0.35 mm					9 (39.1)
Yes, lead equivalence 0.5 mm					4 (17.4)
How often do you wear a cap?	12 (13.2)	5 (5.5)	10 (11.0)	64 (70.3)	
Do you wear a headband?					
Yes, lead equivalence 0.25 mm					-
Yes, lead equivalence 0.35 mm					2 (2.8)
Yes, lead equivalence 0.5 mm					-
How often do you wear a headband?	2 (2.8)	-	-	70 (97.2)	

* Figures quoted as numbers and percentages are based on the valid data.

**Table 3 ijerph-18-13131-t003:** Information on the availability and use of interventional installations.

Question *	Always	Frequently	Rarely	Never	N (%)
Is there a ceiling-suspended lead acrylic shield in the examination room?					
No					4 (4.0)
Yes					95 (96.0)
How often do you use a ceiling-suspended lead acrylic shield?	78 (78.8)	18 (18.2)	1 (1.0)	2 (2.0)	
Is there a drape under the examination table on one side?					
No					26 (28.0)
Yes					67 (72.0)
Is there a drape under the examination table on both sides?					
No					65 (67.7)
Yes					31 (32.3)
Is there an over-table shield?					
No					15 (15.1)
Yes					84 (84.9)
How often is the over-table shield used?	61 (63.5)	20 (20.8)	2 (2.1)	13 (13.5)	

* Figures quoted as numbers and percentages are based on the valid data.

**Table 4 ijerph-18-13131-t004:** Information provided by study participants on other means of reducing radiation.

Question *	Always	Frequently	Rarely	Never	N (%)
Is a patient apron used?					
No					11 (15.3)
Yes, lead equivalence 0.5 mm					56 (77.8)
Yes, lead equivalence 1.0 mm					5 (6.9)
How often is a patient apron used?	30 (34.1)	33 (37.5)	21 (23.9)	4 (4.5)	
Do you use all the technical means of reducing radiation?	49 (51.6)	45 (47.4)	1 (1.0)	-	
Working practices that reduce radiation?					
3-D mapping, navigation system					3 (23.1)
FORS ^1^ technology, participation in studies					2 (15.4)
Using a long wire to change catheters					1 (7.7)
Regular image fades					1 (7.7)
Tiger catheter, distance					5 (38.5)
Syringe pump					1 (7.7)

* Figures quoted as numbers and percentages are based on the valid data. ^1^–FORS = Fiber Optic RealShape.

**Table 5 ijerph-18-13131-t005:** Parameters affecting exposure, such as fluoroscopy time, image frequency and images per examination.

Parameters Affecting Exposure	Average	Standard Error	Standard Deviation	Median	Interquartile Range	Minimum	Maximum	N
Fluoroscopy time in minutes	7.36	0.95	6.53	5	8	1	30	47
Image frequency per second	8.24	0.41	2.93	7.5	2	3	15	52
Images per examination	7.93	0.56	3.88	6	2	1	20	48

## Data Availability

The data set is available upon request. Please send mail to corresponding author.

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
