# Peer review of "Radiation Protection in Interventional Radiology/Cardiology—Is State-of-the-Art Equipment Used?"

_ijerph, 2021, doi:10.3390/ijerph182413131_

Round 1
Reviewer 1 Report
- The reviewer still highly recommend to revise the words of interventional medicine into interventional radiology/Cardiology, as non-ionizing radiation is also used in the interventional medicine.
- The review is still interested in the number of surveyed clinics. The answer of nine is not consistent with the number of eight written in the revised manuscript. Anyway, it is highly recommended to add a limitation of this study in the section of discussions, as the surveyed number of clinics is too small by compared with the total number in Germany.
- References are needed to proof the descriptions in Lines 74-75. In general, using the lead apron over the patients may increase the scattered radiation.
- Please keep in mind that a thickness of more than 0.75 mm lead glasses is usually unnecessary in the radiation field. Please refer to the reference with the DOI:10.1093/rpd/ncw098.
- Also please keep in mind that eyelen dosimeters have been used for many years, they are usually better than other dosimeters over the aprons.
Author Response
Response to Reviewer 1 Comments
Point 1: The reviewer still highly recommend to revise the words of interventional medicine into interventional radiology/Cardiology, as non-ionizing radiation is also used in the interventional medicine.
Response 1: The authors followed the expert's recommendation and revised the words from interventional medicine to interventional radiology / cardiology.
Point 2: The review is still interested in the number of surveyed clinics. The answer of nine is not consistent with the number of eight written in the revised manuscript. Anyway, it is highly recommended to add a limitation of this study in the section of discussions, as the surveyed number of clinics is too small by compared with the total number in Germany.
Response 2: Thank you for pointing this out. In this study, participants from eight German clinics were interviewed. In addition, we added in the limitation part, that the participating clinics were not selected randomly. Therefore, generalization of the results might be problematic (289-290).
Point 3: References are needed to proof the descriptions in Lines 74-75. In general, using the lead apron over the patients may increase the scattered radiation.
Response 3: The following reference is added to the statement in lines 74-75 (76-78): King et al., 2002. In addition we added in line 74 (78): “The use of a patient apron has to be placed outside the field of the primary beam.”
Point 4: Please keep in mind that a thickness of more than 0.75 mm lead glasses is usually unnecessary in the radiation field. Please refer to the reference with the DOI:10.1093/rpd/ncw098.
Response 4: The note that lead glasses with a thickness of more than 0.75 mm in the radiation area are generally not required now is included in the discussion section with the proposed reference: DOI: 10.1093 / rpd / ncw098.
In addition we added in line 267-269: “In other studies, glasses with a lead equivalent of 0.5 mm combined with a large lens are recommended by interventional radiologists as a more adequate and effective protection of the eye lens.”
Point 5: Also please keep in mind that eyelen dosimeters have been used for many years, they are usually better than other dosimeters over the aprons.
Response 5: We recommend in our study that the estimate of the organ-specific equivalent dose for the lens of the eye and the head should be measured at least temporarily over the apron.

Reviewer 2 Report
I was OK with this submission to begin with and so I am content to recommend publication
Author Response
Response to Reviewer 2 Comments
Thank you very much for your positive feedback.